# Mechanistic understanding of in vivo protein corona formation on polymeric nanoparticles and impact on pharmacokinetics

Nicolas Bertrand[1,2], Philippe Grenier[2], Morteza Mahmoudi [3], Eliana M. Lima [1,4], Eric A. Appel [1,5], Flavio Dormont[1], Jong-Min Lim[6,7], Rohit Karnik [6], Robert Langer[1,8] & Omid C. Farokhzad[3,9]

In vitro incubation of nanomaterials with plasma offer insights on biological interactions, but cannot fully explain the in vivo fate of nanomaterials. Here, we use a library of polymer nanoparticles to show how physicochemical characteristics influence blood circulation and early distribution. For particles with different diameters, surface hydrophilicity appears to mediate early clearance. Densities above a critical value of approximately 20 poly(ethylene glycol) chains (MW 5 kDa) per 100 nm$^2$ prolong circulation times, irrespective of size. In knockout mice, clearance mechanisms are identified for nanoparticles with low and high steric protection. Studies in animals deficient in the C3 protein showed that complement activation could not explain differences in the clearance of nanoparticles. In nanoparticles with low poly(ethylene glycol) coverage, adsorption of apolipoproteins can prolong circulation times. In parallel, the low-density-lipoprotein receptor plays a predominant role in the clearance of nanoparticles, irrespective of poly(ethylene glycol) density. These results further our understanding of nanopharmacology.

[1] David H. Koch Institute for Integrative Cancer Research, Massachusetts Institute of Technology (MIT), 500 Main Street, Building 76-661, Cambridge, MA 02139, USA. [2] Faculty of Pharmacy, CHU de Quebec Research Center, Université Laval, 2705 Laurier Blvd, Québec, Canada G1V 4G2. [3] Center for Nanomedicine and Department of Anesthesiology, Brigham and Women's Hospital, Harvard Medical School, 60 Fenwood Road, Boston, MA 02115, USA. [4] Laboratory of Pharmaceutical Technology, Federal University of Goiás, Goiânia 74605-220 Goiás, Brazil. [5] Department of Materials Science & Engineering, Stanford University, 496 Lomita Mall, Stanford, CA 94305, USA. [6] Department of Mechanical Engineering, Massachusetts Institute of Technology, Cambridge, MA 02139, USA. [7] Department of Chemical Engineering, Soonchunhyang University, 22 Soonchunhyang-ro, Shinchang-myeon, Asan-si, Chungcheongnam-do 31538, Korea. [8] Harvard-MIT Division of Health Sciences and Technology, and Department of Chemical Engineering, MIT, Cambridge, MA 02139, USA. [9] King Abdulaziz University, Jeddah 21589, Saudi Arabia. Correspondence and requests for materials should be addressed to O.C.F. (email: ofarokhzad@partners.org)

Upon dilution in plasma, nanoparticles rapidly adsorb proteins[1]. It is believed that this protein corona affects how nanoparticles are perceived by biological systems[2, 3]. In an elegant proteomics study performed on uncoated silica and poly(styrene) nanoparticles, Tenzer et al.[1] showed that equilibrium in the protein corona is reached a few minutes after dilution in plasma. Furthermore, they showed that this corona affects how nanomaterials interact with platelets and blood cells. In biological systems, interactions with proteins ensure adequate distribution of nanoparticles to the desired target. For some siRNA-loaded lipoplexes, interactions with apolipoproteins are essential for the targeting of hepatocytes[4, 5]. In other cases, protein adsorption on targeted silica nanoparticles induces a loss of selectivity in receptor-mediated endocytosis[6]. In clinical settings, interactions of nanomedicines with proteins, notably via the activation of the complement system, can trigger pseudo-allergic reactions[7]. For all these reasons, understanding how nanoparticles interact with the biological milieu is crucial for the rational development of drug-delivery systems.

Nanoparticles prepared with poly(ethylene glycol)-b-poly(lactic co glycolic acid) (PEG–PLGA) copolymers have a long history in drug delivery[8] and have recently reached clinical maturity[9]. In light of the data obtained with model nanoparticles[1, 6, 10, 11], it is therefore particularly interesting to understand how physicochemical properties affect the biological fate of these clinically relevant particles.

Here, we show that the PEG density on the surface of PEG–PLGA nanoparticles is a key determinant of their early clearance in vivo. We identify a PEG density threshold below which blood clearance is rapid. Further PEGylation beyond this value does not significantly prolong the circulation times measured over a period of 6 h. This PEG density threshold, measured as the number of PEG chains per 100 nm², remains similar for nanoparticles with diameters of 55, 90, and 140 nm.

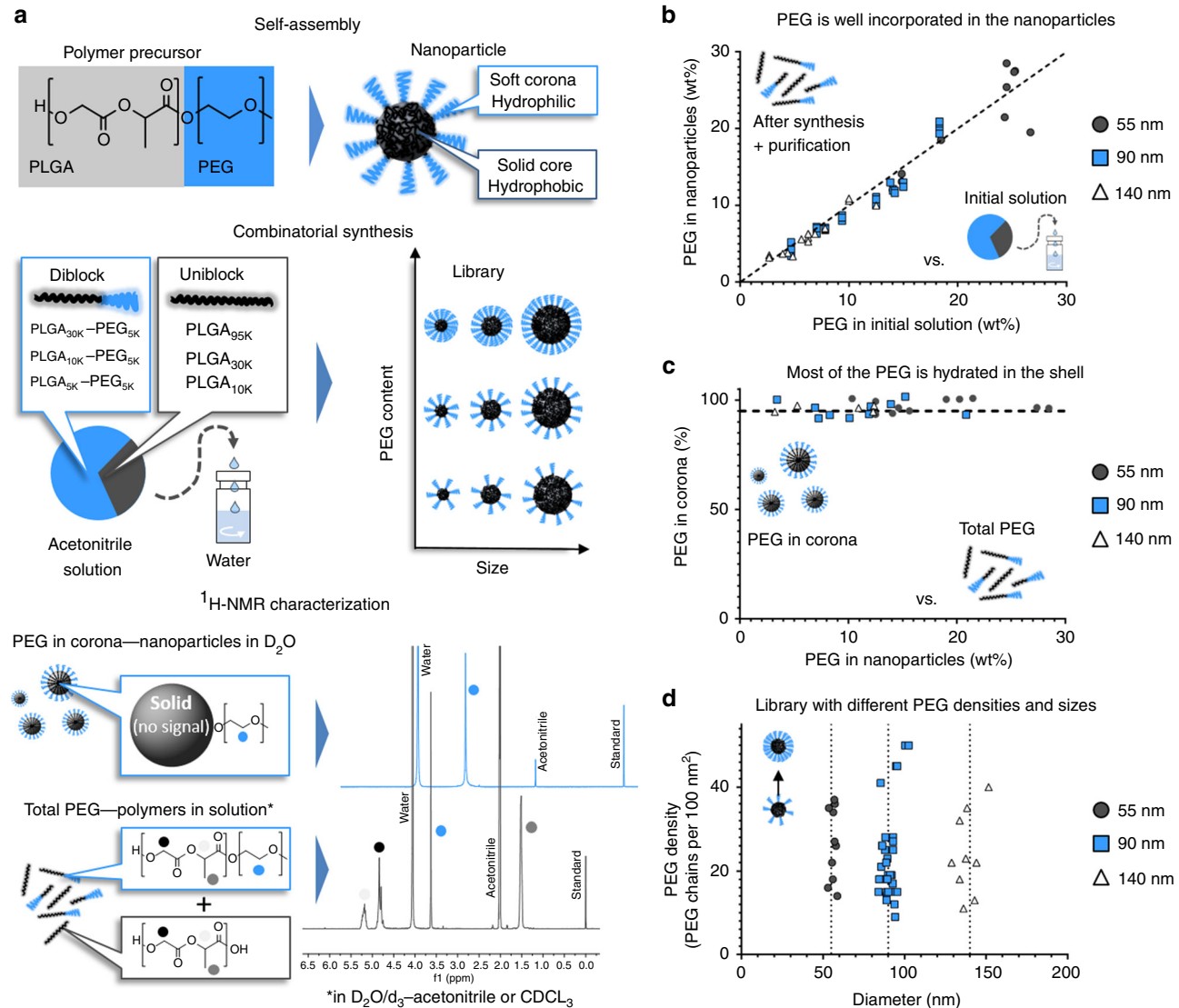

**Fig. 1** Nanoparticles with different PEG densities and sizes prepared by combinatorial synthesis of PEG–PLGA and PLGA copolymers. **a** By self-assembly, polymer precursors form core-shell nanoparticles. Different sizes and PEG contents can be obtained by combining diblock and uniblock precursors. Analysis by ¹H-NMR spectroscopy in different solvents can determine the PEG content in the nanoparticles and in the hydrated shell. **b** Most of the PEG used in the polymer precursor solution is incorporated in the nanoparticles. **c** For nanoparticles with diameters of 55, 90, and 140 nm, whose core is non-solvated (i.e., in D₂O), most of the total PEG signal is detectable, suggesting that most of the hydrophilic polymer is hydrated in the shell. **d** Joining combinatorial synthesis and careful ¹H-NMR characterization, a library of nanoparticles with different sizes and PEG densities can be prepared

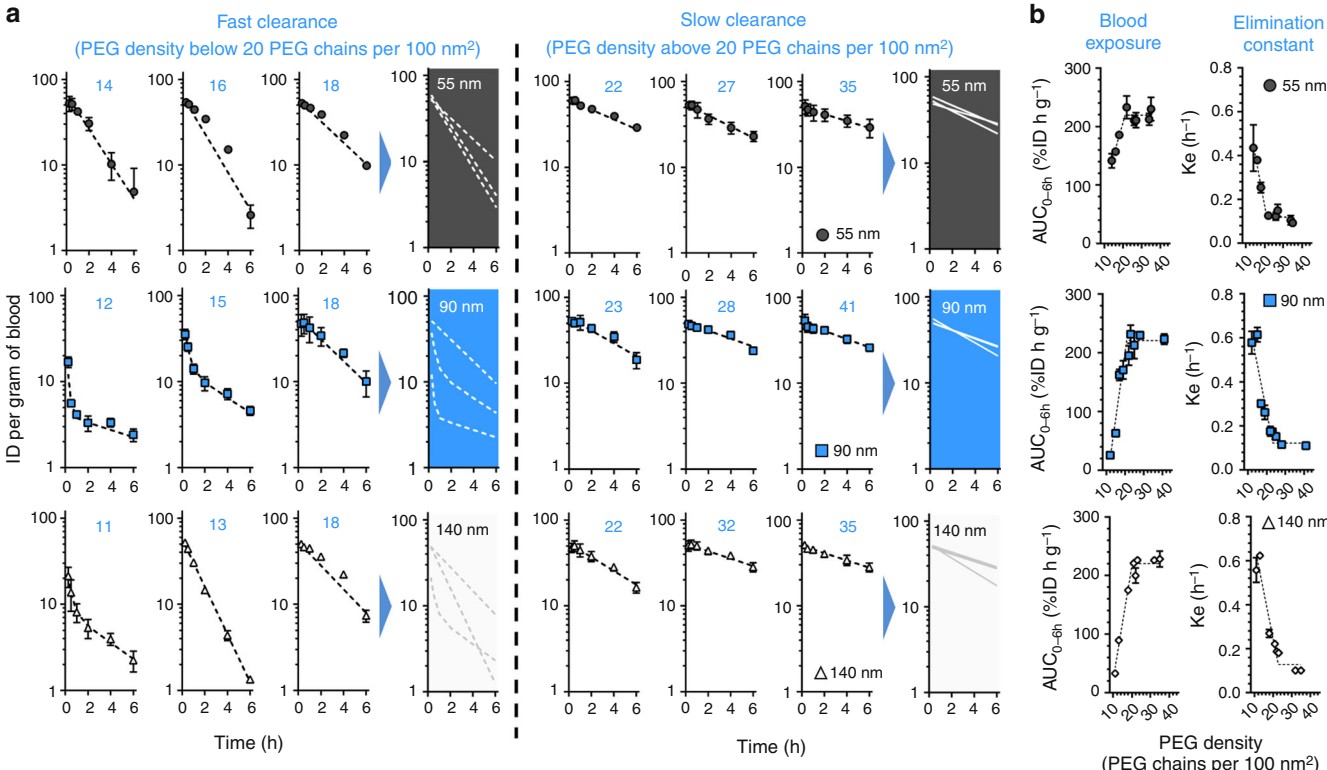

**Fig. 2** A PEG density threshold of 20 PEG chains per 100 nm$^2$ is necessary to avoid early clearance from the bloodstream. **a** The early circulation of nanoparticles with diameters of 55, 90, and 140 nm appears to be mainly affected by the density of PEG chains on their surface. Below a threshold around 20 PEG chains per 100 nm$^2$, the nanoparticles are cleared rapidly; above this value, the nanoparticles exhibit roughly the same circulation profiles, irrespective of their diameter or extent of PEGylation. **b** Pharmacokinetic analysis of blood exposure (AUC$_{0-6h}$) and elimination constants ($k_e$) also highlight the presence of a threshold above which greater PEG coverage does not increase benefits on prolonging the circulation of nanoparticles. Values are means ± SD ($n = 3$-5); numbers in *blue* represent the PEG chains per 100 nm$^2$ of surface

Using transgenic mice, and animals with different protein phenotypes, we highlight that complement activation cannot explain the differences observed between nanoparticles with fast and slow clearances. Interestingly, apolipoproteins responsible for the trafficking of lipids in the bloodstream interact with nanoparticles and appear to impact on their clearance.

## Results

**Nanoparticle synthesis and characterization.** Using nanoprecipitation and combinatorial synthesis[12], PEG–PLGA nanoparticles can be synthesized with different physicochemical properties in a robust and reproducible manner. Furthermore, because they are prepared using structurally simple copolymers, in the absence of other surfactants, proton nuclear magnetic resonance ($^1$H-NMR) can be used to monitor their composition and quantitatively characterize their outside polymer shell[13, 14]. PEG contents assessed by $^1$H-NMR were in good agreement with those measured by a colorimetric, iodine-based PEG quantification method (Supplementary Fig. 1). Herein, PEG–PLGA nanoparticles with different sizes (55–140 nm) and PEG densities (10–50 PEG chains per 100 nm$^2$) were synthesized through nanoprecipitation of PLGA and PEG–PLGA copolymers (Fig. 1 and Supplementary Table 2). A PEG molecular weight of 5000 was chosen, in accordance with clinically advanced PEG–PLGA nanoparticles[9] and other systems that appear to necessitate longer PEG chains than lipid-based platforms[15–17]. PEG content after purification was mostly proportional to the amounts of PEG used in the initial solution, and more than 90% of the total PEG was found solvated in the outside shell (Fig. 1b, c). Using the nanoprecipitation method, it was therefore possible to prepare

particles with nearly neutral zeta potential (between 0 and −5 mV), differing only in size or PEG density (Fig. 1d, and Supplementary Fig. 2). To facilitate their detection in vivo, these nanoparticles were also labeled with small amounts of [$^{14}$C]-PLGA.

**Circulation in wild-type animals.** The circulation profile of [$^{14}$C]-labeled nanoparticles was monitored for 6 h following intravenous injection to healthy Balb/c mice[18, 19]. In agreement with the general consensus in the literature[20, 21], for particles of the same size, greater PEGylation decreases early clearance from the bloodstream (Fig. 2). Interestingly, within that 6-h time frame, little benefit is gained from further increasing the amount of PEG in the system, once the polymer shell reaches around 20 PEG chains per 100 nm$^2$. In other words, two clearance rates are observed: fast and slow, for nanoparticles with densities below and above 20 PEG chains per 100 nm$^2$, respectively. Distribution to the liver and spleen is also different between nanoparticles with PEG densities above and below that threshold (Supplementary Fig. 3). Since the clearance mechanisms of Balb/c mice might be different from those of other animal strains[22], this critical PEG density value was also confirmed in C57Bl/6 mice and rats (Supplementary Figs. 4 and 5), confirming the consistency of these observations across multiple preclinical models. Regarding parameters commonly used to describe PEG density[20, 21], the threshold of 20 PEG chains per 100 nm$^2$ is equivalent to a distance between PEG$_{5k}$ chains of 2.5 nm and a PEG layer thickness of 10.6 nm. These values correspond to a dense PEG brush conformation and appear consistent with the literature on long-circulating liposomes (Supplementary Discussion), but seem

much lower than the PEG density required to obtain long-circulating poly(styrene) nanoparticles[20]. These discrepancies could be explained by differences in the hydrophobicity of the core, or by the methodologies used to quantify the PEG contents.

Compellingly, for particles with diameters of 55, 90, and 140 nm, the threshold of PEG density at which clearance slows down appears to be the same (i.e., around 20 PEG chains per 100 nm[2]). In addition, during this relatively short study, particles with PEG densities above this critical value exhibit very

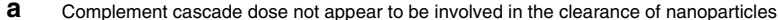

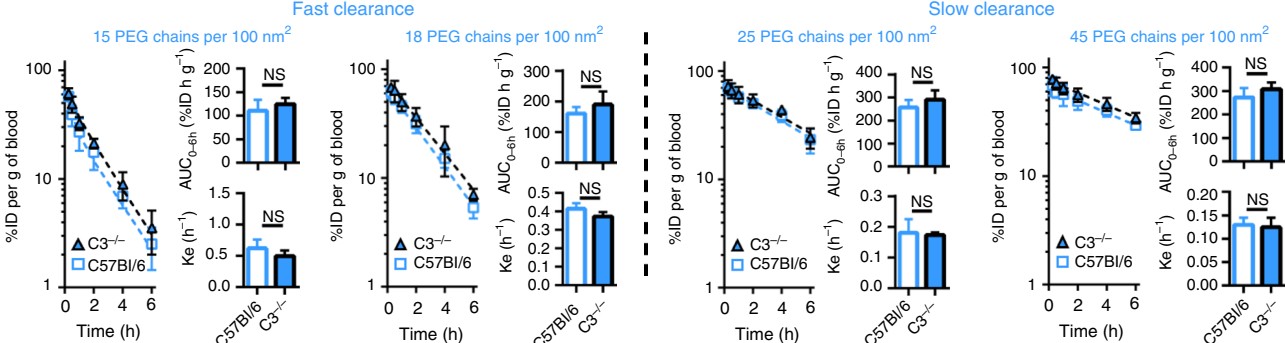

**a** Complement cascade dose not appear to be involved in the clearance of nanoparticles

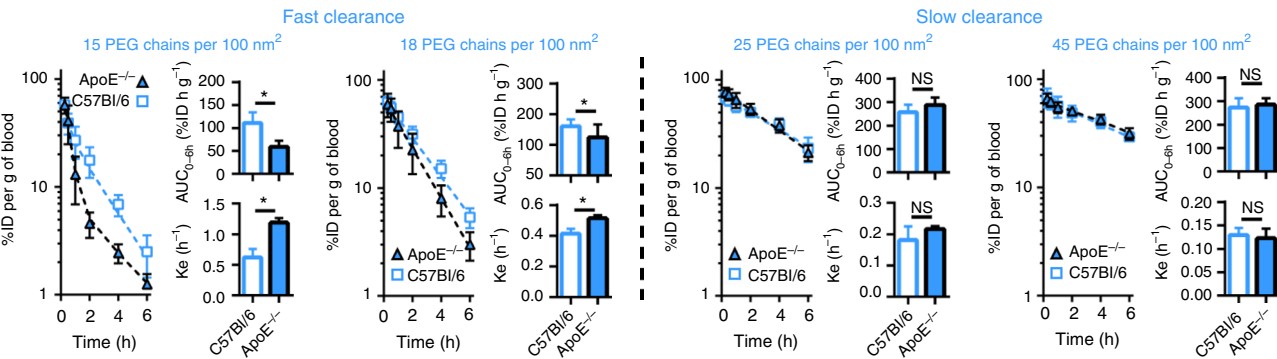

**b** ApoE acts as a dysoponin when PEG density is low, but not when it is high

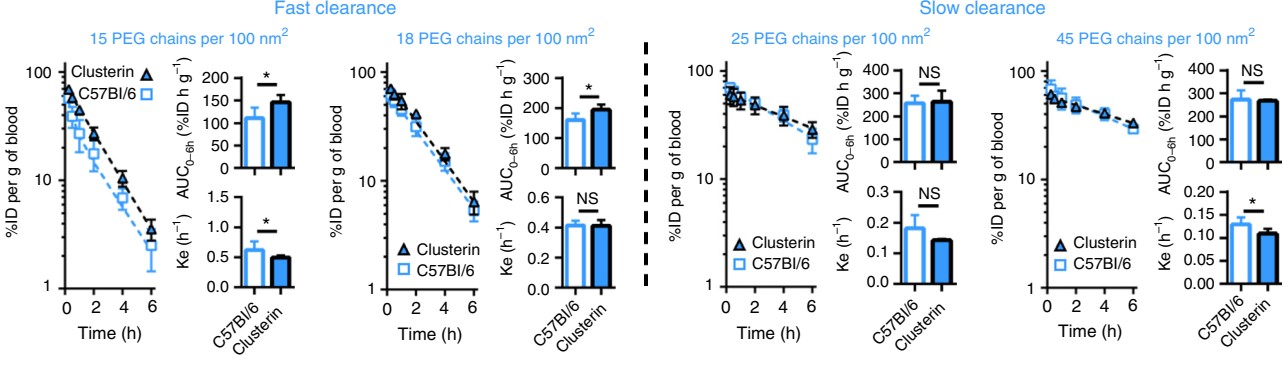

**c** Clusterin acts as a dysopsonin when PEG density is low, but not when it is high

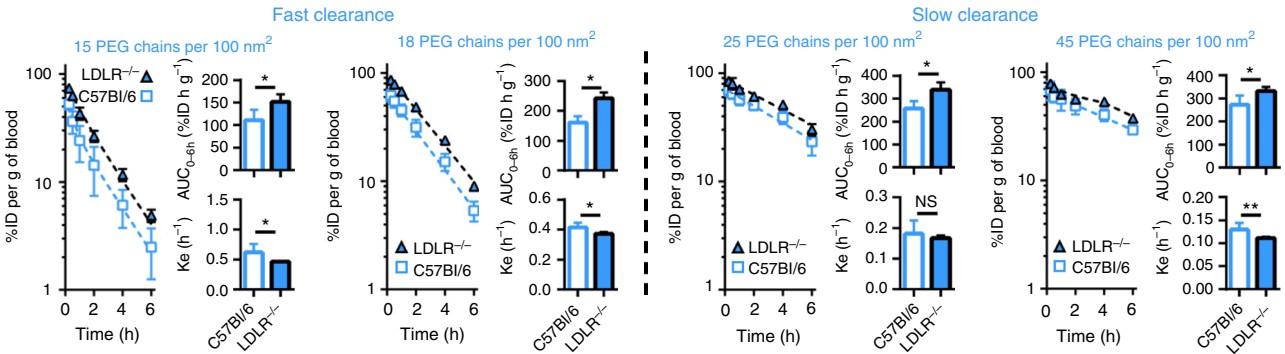

**d** Levels of LDLR influence clearance of nanoparticles

similar circulation profiles; that is, similar area under the blood concentration vs. time curve ($AUC_{0-6h}$) and elimination constants ($k_e$) (Fig. 2, Supplementary Figs. 4 and 5). This suggests that the surface makeup of the particles, and not their dimension, is responsible for their removal from the bloodstream. This observation contrast with data obtained with PEGylated gold and polystyrene particles of different sizes (the former being coated with $PEG_{1k}$, $PEG_{2k}$, and $PEG_{10k}$[23] and the latter by physically adsorbing tocopheryl-$PEG_{1k}$-succinate[24]) as well as non-PEGylated gold and chitosan nanoparticles[25, 26]. Differences in methods, material properties, and preparation strategies, notably regarding PEGylation (Supplementary Discussion) or possible differences in shell stability in vivo[27], might explain these variations. The size range studied here (55–140 nm) remains significantly larger than that of adsorbing proteins and comparable to the diameters of many systems engineered for drug delivery and imaging applications.

**In vivo protein corona.** In many reports, the protein corona formed on metal[28], lipid[29], and polymer nanoparticles[11] appears generally composed of complement proteins and apolipoproteins with or without immunoglobulins. In a comprehensive review on the subject, Monopoli et al. refer to the presence of soft and hard protein coronas[2], i.e., proteins that are loosely or tightly associated with the nanoparticles, respectively. Because only strong interactions can withstand purification processes, the hard corona is likely easier to characterize experimentally[2]. Here, the hard corona formed in the bloodstream on 90-nm nanoparticles within 15 min of their intravenous injection was characterized (particles with 15, 18, 25, and 45 PEG chains per 100 nm$^2$, Supplementary Figs. 6 and 7). In this experiment, the relative abundance of protein compared to plasma, and not their percentage distribution in the corona layer, was analyzed by Tandem Mass Tag (TMT)-label proteomics analysis (Supplementary Method). On the surface of nanoparticles with slow and fast clearance alike, various apolipoproteins were significantly enriched, a phenomenon observed by others[2]. Although no obvious dichotomous differences were found in the proteomic footprint of nanoparticles with fast and slow clearance, the adsorption of apolipoprotein E (ApoE) appeared to be dependent on PEG density (Supplementary Fig. 7). For this protein, nanoparticles with higher PEG densities appear to have lower relative abundance of ApoE.

**Effect of complement activation.** Beyond what can be observed through proteomics analysis, in vivo experiments can also offer insights on how protein interactions affect the overall biological fate of nanoparticles. For example, the blood exposure of nanoparticles with fast and slow clearances can be compared among animals with different phenotypes in blood proteins. Like other nanomaterials, PEGylated PLGA nanoparticles are mild complement activators in vitro, irrespective of their PEG density (Supplementary Fig. 8)[30]. However, upon intravenous injection, they fail to markedly elevate the circulating levels of C5b-9, the end product of the complement cascade (Supplementary Fig. 9). To elucidate whether the complement cascade can partly explain the large differences in circulation times observed with nanoparticles with fast and slow clearance, the blood circulation profiles of animals deficient in complement protein 3 ($C3^{-/-}$) were compared to those obtained in wild-type mice (C57Bl/6) (Fig. 3a). The C3 protein is essential to all three complement activation pathways; these double-knockout animals have undetectable levels of C3 protein and no residual functional complement activity, according to ELISA and hemolytic assays, respectively[31]. In these mice, no increase in circulation times were observed when the cascade is abrogated, suggesting that this part of the innate immunity cannot explain the significant differences in the clearance of nanoparticles with PEG densities above and below 20 PEG chains per 100 nm$^2$ (Fig. 3a). Correspondingly, similar amounts of C3 and other complement components were measured by proteomics on nanoparticles with fast and slow clearances (Supplementary Fig. 6c). Together, this suggests that the deposition of complement proteins on the surface of PEG–PLGA nanoparticles is too mild to impact their circulation times, irrespective of their PEG density. PEGylated liposomes are also mild complement activators: despite their very long circulation times[32], they can trigger pseudo-allergic infusion reactions by activating the cascade[7, 33]. In contrast, Wang et al.[34, 35] showed that the cellular distribution of strong complement activators within circulating phagocytes could be impacted by the activation of the cascade.

**Effect of lipid trafficking pathways.** In parallel, the pathways involved in lipid trafficking also appear to play important roles in the clearance of nanoparticles, in accordance with the noticeable enrichment of apolipoproteins (in terms of their relative abundance compared to plasma) in the hard corona, after exposure to in vivo conditions. Because the apolipoprotein is undetectable in the plasma of animals deficient in ApoE (ApoE$^{-/-}$)[36], the model can therefore be used to monitor the impact of this specific protein on nanoparticle circulation. Here, the presence of ApoE appears to mitigate the clearance of nanoparticles with low PEG coverage. The depletion of ApoE resulted in the accelerated clearance of nanoparticles with low PEG densities (i.e., a 20–30% decrease in $AUC_{0-6h}$ and 20–60% increase in $k_e$ for nanoparticles with 15 and 18 PEG chains per 100 nm$^2$, Fig. 3b). This effect was not observed for nanoparticles with higher PEG coverage.

ApoE is among the most abundant proteins detected in the proteomics experiment, and appears to preferentially interact with the surface of the nanoparticles (Supplementary Fig. 6). In knockout animals, the absence this protein has a substantial impact on the protein corona, and favors faster blood clearance, conceivably due to a higher relative contribution of opsonins. The effect is potentially more visible with nanoparticles with low-steric protection, given that they intrinsically adsorb higher quantities of ApoE (Supplementary Fig. 7). To assess this hypothesis, nanoparticles with 15 PEG chains per 100 nm$^2$ were

**Fig. 3** Circulation profiles in different mouse models highlight the role of different proteins on the clearance of nanoparticles. **a** For all nanoparticles, the circulation profiles between wildtype and C3$^{-/-}$ animals are similar. This suggests that the complement cascade is not involved in the clearance of nanoparticles, even those with very low steric protection and fast clearance. **b** The absence of ApoE accelerates the clearance of nanoparticles with fast intrinsic clearance. This suggests that when steric protection is low, interactions with ApoE prevent clearance-enhancing proteins from adsorbing on the nanoparticles. This effect is not observable for nanoparticles with higher PEG densities. **c** Similar to ApoE, pre-adsorption of clusterin on the surface of nanoparticles with low PEG densities decreases clearance. Clusterin does not appear to influence nanoparticles with slower intrinsic clearance rates. **d** In LDLR$^{-/-}$ animals circulation times are prolonged, with augmented blood exposures for all nanoparticles. This suggests direct involvement of LDLR on the clearance of nanoparticles. Values are means ± SD ($n = 4$–13). *$p < 0.05$ as determined by t-test, **$p < 0.05$ as determined by Mann–Whitney (non-parametric) test

incubated in plasma from ApoE$^{-/-}$ and control animals. When these pre-opsonized nanoparticles were injected into C57Bl/6 mice, nanoparticles coated with plasma from ApoE$^{-/-}$ animals had a 1.3-fold lower blood exposure compared to those coated with normal plasma (Supplementary Fig. 10). This further supports the idea that, for nanoparticles with low PEG densities, ApoE plays a dysopsonic role. It should be emphasized that additional mechanisms may also be involved in the changes observed in vivo, as the elimination of ApoE protein may have multiple downstream effects and can change other proteins in plasma as well. In comparison, in nanoparticles with higher PEG density and slower intrinsic clearance, ApoE represents a lower relative abundance in the protein corona; the changes in the corona composition after depleting ApoE are therefore comparatively smaller.

Recently, Schöttler et al. elegantly highlighted the possible role of another apolipoprotein, clusterin (also known as ApoJ), in preventing the uptake of sterically protected nanoparticles by macrophages[37]. In the current study, pre-incubation with clusterin before injection increased the in vivo blood exposure of nanoparticles with fast intrinsic clearance (Fig. 3c). However, the enrichment of clusterin on the surface of nanoparticles with higher PEG densities did not significantly alter their blood circulation (Fig. 3c). Similar to ApoE, clusterin appears to shield nanoparticles with insufficient PEG coverage against opsonization, but fails to affect the circulation of nanoparticles that have greater constitutive steric protection. However, one must note that in this experimental setting, only fairly strong interactions would prevent exogenous proteins from desorbing from the nanoparticle surface upon contact with the bloodstream. To that point, Chen et al.[38] showed that the complement factor C3 could desorb from the surface of nanoparticles, despite being covalently attached to the nanoparticle corona.

The effect of ApoE on nanoparticles with low PEG densities contrasts with the results obtained in mice deficient in the LDL receptor (LDLR$^{-/-}$). Like other knockout animals, these mice show undetectable hepatic levels of LDLR[39]. Depletion of the cellular receptor resulted in lower clearance for all nanoparticles, with a 1.2-fold to 1.7-fold increase in AUC$_{0-6h}$ and a 1.1-fold to 1.6-fold decrease in $k_e$ (Fig. 3d). Physiologically, LDLR is a cellular protein responsible for receptor-mediated endocytosis of lipoproteins coated with ApoE and ApoB100[40]. To assess whether ApoB100 affects clearance, nanoparticles were incubated in plasma from LDLR$^{-/-}$ ApoB100$^{only}$ mice, which have 10-fold higher ApoB100 levels than wild-type animals[41]. For all nanoparticles tested, the blood circulation profiles remained similar to the controls (Supplementary Fig. 10), suggesting that the impact of ApoB100 on nanoparticle clearance is minimal. This is not surprising in light of the lower intrinsic affinity of that protein for LDLR[39].

Proprotein convertase subtilisin/kexin type 9 (PCSK9) is another physiological ligand of LDLR that downregulates the receptors by triggering their internalization and trafficking to lysosomal vesicles[42]. Within 60 min of intravenous injection, small doses of PCSK9 transiently reduce the levels of LDLR in the liver by 80%[42]. Here, pre-dosing animals with 16 µg of recombinant mouse PCSK9 before the administration of nanoparticles confirmed the role of LDLR receptors in nanoparticle clearance: animals receiving the recombinant protein showed longer nanoparticle circulation times compared to untreated wild-type animals (1.2 to 1.7-fold higher AUC$_{0-6h}$ and 1.1–1.4 lower $k_e$) (Supplementary Fig. 11). Similar to the results obtained in LDLR$^{-/-}$ mice, this effect was noticeable for all 90-nm nanoparticles, irrespective of PEG density. One dose of PCSK9, given 60 min before the injection of nanoparticles, could hardly alter the lipid profile of wild-type animals and significantly

change the interactions between proteins and nanoparticles. These findings therefore strongly support the direct involvement of interactions with LDLR in the clearance of nanoparticles from the bloodstream.

## Discussion

In the current study, a library of clinically relevant nanoparticles allowed the identification of an effective threshold of PEG density, which appears to toggle between fast and slow clearance over a broad range of diameters. Combinatorial synthesis and methodical $^1$H-NMR characterization provide blueprints to engineer nanoparticles with different diameters but similar circulation times, at least during the six first hours following injection. The availability of such systems might stimulate many fundamental studies on the biological fate of nanoparticles. Our findings also confirm that surface properties are very important determinants of early interactions with the host's defense mechanisms, arguably more than size, when nanoparticles are initially introduced into the bloodstream. Even within the relatively short time frame studied, blood exposure varied up to 7-fold between short-circulating and long-circulating nanoparticles. Given the complexity of clearance mechanisms, in vivo pharmacokinetics proved an efficient tool to highlight these differences and supplement other work elucidating the complex interactions at the nano-bio interface[1, 6]. Animals with different protein phenotypes and knockout mouse models prove valuable to study the role of the in vivo protein adsorption on the removal of nanoparticles from the bloodstream. In that context, ApoE exhibited distinct functions on nanoparticles with low and high PEG densities. While it appears to protect nanoparticles with poor steric protection against rapid opsonization upon entry into the bloodstream, it also seems to act as a potential ligand for LDLR on all nanoparticles. The involvement of LDLR in the clearance of nanoparticles was confirmed using two different models. Despite the fact that ApoE is likely involved in these interactions, it still remains unclear whether the presence of the apolipoprotein is totally indispensable. As novel therapeutic platforms are developed to treat cardiovascular diseases[43], the impact of serum lipids on the clearance of nanoparticles might become increasingly important. In parallel, our findings in complement-deficient animals shed light on the relevance of this cascade for PEGylated nanoparticles, at least from the perspective of clearance in naive mice. Our results suggest that complement activation cannot be the sole predictor of circulation times in mice given that the biological fate of both short-circulating and long-circulating nanoparticles appeared unaffected by disruption of this cascade. The biological relevance of the complex interactions between nanoparticles and this cascade of the innate immunity are still being unraveled[38], especially in mice where instability of certain activation pathways has been highlighted[35]. Further studies are also required to understand the role of this cascade with regard to clearance after multiple doses of nanoparticles[44] or with regard to possible pseudo-allergic reactions in patients[45]. Overall, a better understanding of the clearance mechanisms of nanoparticles and improved control over their circulation times might help engineer more efficacious systems, whether they are utilized for in vivo imaging[46], tumor targeting[47], or other biomedical applications[48].

## Methods

**Polymer synthesis**. PEG–PLGA copolymers were synthesized by a semi-batch ring-opening polymerization of D,L-lactide and glycolide at room temperature using mPEG$_{5k}$-OH as an initiator and 1,8-diazabicycloundec-7-ene (DBU) as a catalyst[49]. In a typical polymerization, after drying the initiator and monomers under vacuum overnight, mPEG$_{5k}$-OH (0.134 mmol) and D,L-lactide (10.32 mmol) were solubilized in ~40 ml anhydrous dichloromethane. In parallel, a solution of glycolide (5.16 mmol) was prepared using 7 ml anhydrous tetrahydrofuran (THF).

Upon initiation of the polymerization by introduction of DBU (0.134 mmol), the glycolide solution was immediately added at a rate of 0.7 ml min$^{-1}$ using a syringe pump. After 10 min, the polymerization was stopped by addition of benzoic acid (1.5 mmol). The solvent was removed by rotary evaporation, and the polymer was precipitated twice in cold diethyl ether and dried over vacuum. $^1$H-NMR (CDCl$_3$, 400 MHz): δ 1.58 p.p.m. (159 H, C(CH$_3$)H), 3.37 p.p.m. (3 H, H$_3$COCH$_2$CH$_2$), 3.64 p.p.m. (444 H, OCH$_2$CH$_2$), 4.82 p.p.m (114 H, OCH$_2$CO), 5.19 p.p.m. (53 H, OCH(CH$_3$)CO). The polymers used in the preparation of the nanoparticles are presented in Supplementary Table 2.

**Nanoparticle preparation**. Nanoparticles were prepared by nanoprecipitation from acetonitrile solutions[12]. Briefly, polymer precursor solutions were mixed at different ratios and added dropwise to 8–12 ml of water under stirring. The polymer concentration, stirring speed, and volume of water were modified to obtain particles of the desired size. To enable tracking of nanoparticles in vivo, small quantities of $^{14}$C-labeled PLGA polymer (Mn around 20,000, Moravek Biochemicals) were integrated into the polymer mixtures. For the preparation of 55-nm nanoparticles with low PEG density, a microfluidic rapid nanoprecipitation method was used[19, 50]. Nanoparticles were purified and washed with water at least four times using an ultrafiltration device (molecular weight cut-off 100,000) and filtered on a 0.22 μm filter before injection. The nanoparticles' size (Z-average), size distribution (polydispersity index), and zeta potential were measured before and after purification by dynamic light scattering at 22 °C with a 173 backscatter angle, using a Malvern Zetasizer Nano ZS (Malvern Instruments, Westborough, MA).

**$^1$H-NMR characterization of the nanoparticles**. The PEG content in the nanoparticle and in the outside shell were determined by $^1$H-NMR spectroscopy by modification of a method described elsewhere[13]. Generally, the methylene protons of PEG (3.6 p.p.m.) were quantitatively compared to the protons of the lactic (1.6 and 5.2 p.p.m.) and glycolic repeating units (4.8 p.p.m.) to determine how much PEG was present in the blends forming the nanoparticles. To determine the amount of PEG in the shell, a first reading was done in D$_2$O and compared to the spectra obtained in D$_2$O/d$_3$-ACN solvent mixtures. In the former, only the PEG protons (3.6 p.p.m) are observed, while in the latter, the whole polymer being soluble, all protons are visible. To compare spectra, 1 wt% trimethylsilyl propanoic acid (TMSP) was used as an internal standard. The density of the PEG shell was calculated as described elsewhere[14], using the PEG content (percent by mass), a polymer density of 1.2 g cm$^{-3}$, a PEG molecular weight of 5000 g mol$^{-1}$, and the surface (cm$^2$) and volume (cm$^3$) of a nanoparticle calculated from the Z-average. Results from the nanoparticle synthesis and characterization are presented in Supplementary Table 3.

**In vivo studies**. All animal experiments were conducted using institutionally approved protocols at MIT (IACUC) and Université Laval (Canadian Council on Animal Care standards and Animal Research: Reporting In Vivo Experiments guidelines). Healthy animals were housed in a controlled environment (22 °C, 12 h day/night cycle) with ad libitum food and drink access. In a typical experiment, male mice (25–29 g) or Sprague-Dawley rats (220–240 g) were intravenously injected by the subclavian vein under isoflurane anesthesia (2.5 %) with 60 mg kg$^{-1}$ of nanoparticles (20 mg kg$^{-1}$ for rats). During the following 6 h, ~30–50 μl of blood (200–300 μl for rats) was collected via the saphenous vein (0.25, 0.5, 1, 2, 4 h), as well as through a terminal cardiac puncture (6 h). At the end of the experiment, animals were euthanized by a cardiac perfusion of ~3 ml of phosphate-buffered saline solution (pH 7.4, 3 mM phosphate, 150 mM sodium chloride) and organs were collected. Biological samples were digested at 60 °C (Solvable, Perkin Elmer, Waltham, MA), bleached with 30% hydrogen peroxide, and assessed by scintillation counting (Hionic Fluor, Perkin Elmer, Waltham, MA). To assess the protein corona after in vivo exposure, Balb/c mice were injected with nanoparticles with 15, 18, 25, and 45 PEG chains per 100 nm$^2$. Fifteen minutes after injection, the radiolabeled nanoparticles were recovered by cardiac puncture and isolated from the blood using size-exclusion chromatography (Sephacryl S-400 HR) and ultrafiltration (Vivaspin, MWCO 1000 kDa). The amount of protein on the surface of the nanoparticle was quantified using the 660 nm protein quantification assay (Thermo, Waltham, MA).

In specified experiments, nanoparticles were preincubated at 37 °C for 30 min, with EDTA-containing plasma from C57Bl/6 (JAX #027), ApoE$^{-/-}$ (JAX #2052), LDLR$^{-/-}$ ApoB100$^{only}$ (JAX # 3000, kind donation of Dr. Andre Marette, Institut Universitaire de Cardiologie et de Pneumologie de Quebec Research Center) or mouse recombinant clusterin (R&D systems, Minneapolis, MN). The nanoparticle concentration was around 20 mg ml$^{-1}$ in 50% plasma or 100 μg ml$^{-1}$ with protein. Proteins, including mouse recombinant PCSK9 (Speed Biosystems, Gaithersburg, MD), were diluted according to the manufacturer's instructions. In specified experiments, 16 μg of PCSK9 were injected 1 h before nanoparticle administration. The number of animals used in each experiment is presented in Supplementary Table 3.

**Pharmacokinetic parameters**. Non-compartmental analysis of the pharmacokinetic parameters were calculated from blood concentration (%ID per gram of blood) vs. time profiles[48]. The concentration at time 0 ($C_0$) was obtained from the

Y-intercept of linear least-squares regression on the semilog plot of the blood concentration vs. time curve, using the first points of the curve. The volume of distribution (Vd) was obtained from the ratio of the injected dose (100%) over $C_0$. The trapezoidal method, from 0 to 6 h, was used to calculate the AUC$_{0-6h}$ values. The ratio of clearance (dose/AUC$_{0-inf}$) over Vd afforded an estimated elimination constant ($k_e$).

**Statistics**. Statistics were computed with GraphPad Prism 6. Standard unpaired t-test or Mann–Whitney test (non-parametric, when samples failed normality or equality of variance statistical tests) were used to test for statistical significance between groups. A value of $p < 0.05$ was considered significant.

**Data availability**. Proteomics data is deposited on the ProteomeXchange repository (see Supplementary Methods for login details). Supplementary Table 5 presents raw data from the pharmacokinetics experiments. All other data is available from the authors upon reasonable request.

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

## Acknowledgements

This work was supported by the National Cancer Institute (NCI) (grant U54-CA151884), the National Heart, Lung, and Blood Institute (NHLBI) Program of Excellence in Nanotechnology (PEN) (contract #HHSN268201000045C), the National Institute of Biomedical Imaging and Bioengineering (NIBIB) R01 grant (EB015419-01), the David Koch-Prostate Cancer Foundation Award in Nanotherapeutics, and the Koch Institute Support (core) (grant P30-CA14051) from the NCI. N.B. acknowledges a postdoctoral fellowship from the Canadian Institutes of Health Research (CIHR) and funding from the Natural Sciences and Engineering Research Council of Canada (Discovery Grant). EML acknowledges the MIT-Brazil Cooperation Program (grant 202996/2014-0) from CNPq. E.A.A. acknowledges support from a Wellcome Trust–MIT postdoctoral fellowship. We thank the Koch Institute Swanson Biotechnology Center for technical support. We also heartily thank Dr Amanda Del Rosario at the Biopolymers and Proteomics core (Swanson Biotechnology Center), as well as Dr Sylvie Bourassa and Dr Arnaud Droit (both at the Proteomics Core of the CHU de Quebec Research Center) for critical discussion on the proteomics data. We thank Dr Andre Marette (Institut Universitaire de Cardiologie et de Pneumologie de Quebec Research Center) for donating plasma from LDLR$^{-/-}$ ApoB100$^{only}$ mice.

## Author contributions

N.B., R.K., R.L., O.C.F. designed the experiments and wrote the paper. N.B., P.G., E.M.L., E.A.A., F.D., and J.L. conducted the experiments. N.B., M.M., and E.M.L. analyzed the data. R.K., R.L., and O.C.F supervised the research. All authors discussed the progress of research and reviewed the manuscript.

## Additional information

**Supplementary Information** this paper at doi:10.1038/s41467-017-00600-w.

**Competing interests:** O.C.F. and R.L. have financial interests in Tarveda Therapeutics, Selecta Biosciences and Placon Therapeutics. R.L. declares financial interests in Moderna. These biotechnology companies are developing nanoparticle technologies for medical applications. The remaining authors declare no competing financial interests.

