## [Peer Review File · Nature Communications]

Reviewers' comments:

Reviewer #2 (Remarks to the Author):

My comments are focused only on the complement data and conclusions, as the other Reviewers have had issues mainly with other aspects of the study which have been addressed by the authors. I also see that the authors have revised their original conclusion on the suggested impact of complement activation on the clearance of low-density PEG NPs, now saying that there was no difference in circulation time in wild-type and C3-KO mice.

Nevertheless the conclusions regarding the role of complement in PEG-PLGA NP clearance are contradictory and -in my opinion- incorrect. The authors show no C3 binding to these NPs (Suppl Fig. 6C), yet they claim complement activation in wild-type mice, in contrast to C3 KO mice, where it is absent. I do not understand what is the basis of talking about complement activation, when it was not measured in the study by any means?

The conclusion "Strikingly, studies in animal deficient in the C3 protein showed that complement activation does not appear to be involved in the clearance of nanoparticles" is not supported by any data, and by questioning an old paradigm (i.e., that complement activation entails rapid clearance) it may confuse the field without reason. The additional C3 "abrogation" experiments using Balb/c and CVF-treated C57bl/6 mice, that the authors mention in their answer to Reviewer 1, do not add to the question what happens in wild-type animals, so they are irrelevant.

A large number of studies by Ishida et al report complement activation as cause of "accelerated blood clearance" (ABC) of liposomes and other PEGylated nanoparticles in many species, including mice, and recent studies by Moghimi et al report complement activation by PEGylated "nanoworms" closely correlating with macrophage uptake, were these studies neglected? If not, then what is the reason of discrepancy? As for the claim that the authors' results are consistent with the long circulation time of pegylated liposomes despite complement activation is not a strong argument as it can be explained by many other effects, see for example Szebeni et al., J Liposome Res 10:467, 2000.

I would suggest that if the authors wish to maintain their conclusion about the lack of impact of C activation on their NP clearance they should provide evidence of either increased C3 binding to their NPs in wild-type vs KO mice) e.g., proteomics analysis, Suppl Fig 6, or demonstrate C activation, either by the rise of C split products in blood, or signs of C- activation-related symptoms in wild-type mice. The issue of C activation in mice is highly debated in any way, partly because of the known instability of the classical pathway in this species.

It is possible that the authors miss to recognize what is really interesting and important in their study, namely that PEG-PLGA NPs do not bind C3 and activate C simply because they are near neutral. This would explain the identical clearance of these NPs in mice and rats, this would be the real novelty of the paper at least regarding complement.

Reviewer #3 (Remarks to the Author):

The authors have correctly addressed my concerns regarding the presentation and evaluation of the proteomics data. Raw data and search results have been submitted to the ProteomeXchange repository, allowing other researchers to reproduce the analyses.

The authors have correctly re-evaluated the proteomics data and the data presented are now supported by the raw data.

Reviewer #2:

My comments are focused only on the complement data and conclusions, as the other Reviewers have had issues mainly with other aspects of the study which have been addressed by the authors. I also see that the authors have revised their original conclusion on the suggested impact of complement activation on the clearance of low-density PEG NPs, now saying that there was no difference in circulation time in wild-type and C3-KO mice.

Nevertheless the conclusions regarding the role of complement in PEG-PLGA NP clearance are contradictory and -in my opinion- incorrect. The authors show no C3 binding to these NPs (Suppl Fig. 6C), yet they claim complement activation in wild-type mice, in contrast to C3 KO mice, where it is absent.

We thank the reviewer for the comment, and we agree that the previous version of the manuscript might have been unclear in this regard. First, we would like to point out that the Shotgun Proteomics methodology employed to obtain the results presented on Figure 6C is not specifically designed to measure complement activation – 1) shotgun LC/MS/MS proteomics detects (unique) peptide sequences, it cannot discriminate between the C3 full pro-protein and its activated/deactivated forms; and 2) although C3 was one of the proteins found in the highest abundance in the samples (more than 270 spectra and 55 unique peptides), the amounts found in nanoparticle samples were not significantly different from those in control plasma samples. For those reasons, it is difficult to interpret how nanoparticles activate complement solely using shotgun proteomics. That being said, we observed that the amounts of C3 detected in all nanoparticles samples were similar, irrespective of clearance rates and/or PEG density. We amended the main text to highlight these nuances:

In these mice, no increase in circulation times were observed when the cascade is abrogated, suggesting that this part of the innate immunity cannot explain the significant differences in the clearance of nanoparticles with PEG densities above and below 20 PEG chains per 100 nm² (Figure 3a). Correspondingly, similar amounts of C3 and other complement components were measured by proteomics on nanoparticles with fast and slow clearances (Supplementary Figure 6c and Supplementary excel file). Together, this suggests that the deposition of complement proteins on the surface of PLGA-PEG nanoparticles is too mild to impact their circulation times, irrespective of their PEG density. PEGylated liposomes also figure among mild complement activators: despite their very long circulation times,³² they can trigger pseudo-allergic infusion reactions by activating the cascade.^{7, 33} In contrast, Wang and colleagues^{34, 35} showed that the cellular distribution of strong complement activators within circulating phagocytes could be impacted by the activation of the cascade.

I do not understand what is the basis of talking about complement activation, when it was not measured in the study by any means?

We thank the reviewer for the wise comment. It is true that in the previous version, no experiment showed complement activation by our nanoparticles. However, complement activation by PEGylated PLA or PLGA nanoparticles has been shown before (Jablonowski et al. (2016) Biomaterials; D'Addio S et al. (2012) J Control Release; Gaucher et al. (2009) Biomacromolecules; Mosqueira VCF et al. (2001) Biomaterials). Similarly, our group has previously shown that PEGylated PLGA nanoparticles activated complement *in vitro* (Salvador-Morales et al. (2009) Biomaterials).

To further address this reviewer's comment, we carried additional experiments to show that nanoparticles with both high (45 PEG chains per 100 nm²) and low PEG densities (15 PEG chains per 100

nm²) could activate complement *in vitro*, as shown by the production of the C5a anaphylatoxin (Supplementary Figure 8). Interestingly, no obvious differences were perceived on the complement activation of nanoparticles with high and low PEG densities.

Supplementary Figure 1 Nanoparticles with low and high PEG densities (fast and rapid blood clearance, respectively) are mild activators of the complement cascade, irrespective of the PEG density (n = 3, * p < 0.05 as determined by one-way ANOVA and Tukey's *post-hoc*).

To assess whether signs of complement activation could be perceived *in vivo*, nanoparticles were injected to healthy mice, and the concentrations of the terminal complex of the complement cascade, C5b-9, were monitored over time. This experiment showed no significant changes in the circulating levels of C5b-9.

Supplementary Figure 9 Intravenous injections of nanoparticles with low and high PEG densities (fast and rapid blood clearance, respectively) fail to significantly increase the circulating levels of the terminal complex of the complement cascade (C5b-9) in Balb/c mice (n =3).

Overall, this additional data suggests that PEG-PLGA nanoparticles are mild activators of the complement cascade in mice. The main text was modified to convey this idea:

Like other nanomaterials, PEGylated PLGA nanoparticles are mild complement activators *in vitro*, irrespective of their PEG density (Supplementary Figure 8).³⁰ However, upon intravenous injection, they fail to markedly elevate the circulating levels of C5b-9, the end product of the complement cascade (Supplementary Figure 9). To elucidate whether the complement cascade can in part explain the large

differences in circulation times observed with nanoparticles with fast and slow clearance, the blood circulation profiles of animals deficient in complement protein 3 (C3^{-/-}) were compared to those obtained in wildtype mice (C57Bl/6) (Figure 3a).

The conclusion “Strikingly, studies in animal deficient in the C3 protein showed that complement activation does not appear to be involved in the clearance of nanoparticles” is not supported by any data, and by questioning an old paradigm (i.e., that complement activation entails rapid clearance) it may confuse the field without reason. The additional C3 “abrogation” experiments using Balb/c and CVF-treated C57bl/6 mice, that the authors mention in their answer to Reviewer 1, do not add to the question what happens in wild-type animals, so they are irrelevant.

We appreciate the concerns of the referee. However, we would like to point out that, for a range of PLGA-PEG nanoparticles showing a more than 7-fold variation in total blood exposure, no differences is perceived in the blood profiles obtained in WT and C3^{-/-} animals. **Considering these results, it is therefore difficult to suppose that complement activation plays an important role in the clearance of the nanoparticles.** While it is conceivable that the studied PLGA-PEG nanoparticles could be weaker complement activator than other nanomaterials, it seems reasonable to conclude that, for PLGA-PEG nanoparticles with slow and fast clearance alike, abrogation of the complement cascade does not influence the circulation kinetics.

Regarding the “questioning of an old paradigm that complement activation entails rapid clearance”, we would like to point out that this paradigm was already questioned more than 15 years ago with PEGylated liposomal doxorubicin. The referee even correctly refers to a publication that quotes: “Considering that C activation is one of the most powerful mechanism of opsonisation, the finding that Doxil activates C raises the question, how it avoids ready phagocytic uptake? While at present we have no conclusive explanation, we have ruled out at least one possibility: inhibition by PEG-2000 of the formation of iC3b on Doxil (...) (Szebeni et al. (2000) J Liposome Res).” In a more recent publication (Intura et al. (2015) ACS Nano (PMID: 26488074)), Moghimi’s group also mention: “Studies using liposomes bearing PEG 2000 showed that although complement is activated, steric barrier of PEG is strong enough to prevent the binding of C3 opsonized liposomes to macrophages (...) The reasons why crosslinked polymer coating renders nanoparticles less recognizable by (human) leukocytes despite C3 opsonization need to be further investigated (...)”

A large number of studies by Ishida et al report complement activation as cause of "accelerated blood clearance" (ABC) of liposomes and other PEGylated nanoparticles in many species, including mice, and recent studies by Moghimi et al report complement activation by PEGylated "nanoworms" closely correlating with macrophage uptake, were these studies neglected? If not, then what is the reason of discrepancy?

We thank the reviewer for highlighting these publications which allow a higher-level interpretation of our present data. Indeed, complement activation by nanoparticles has been studied for quite some time, and we think our work can provide a new perspective on the subject.

Recent studies by Moghimi’s and Simberg’s groups on nanoworms (Wang et al. (2017) Front Immunol (PMID: 28239384); Inturi et al. (2015) ACS Nano, PMID: 26488074) focus on a different system: SPIO nanoworms coated with 20 kDa dextran (cross-linked or not, decorated with antibodies or not). In their hands, under certain conditions, nanoworms interact with circulating leukocytes in a complement-dependent manner. When comparing these studies with the present work, three things must be taken into consideration:

- 1) The methodologies used in Moghimi’s work and ours differ. To assess interactions between nanoparticles and phagocytes, their technique relies on counting the number of cells containing superparamagnetic objects and isolated from the circulation, 10 minutes, 1 hour or 24 hours after

injection. In our case, we assess the total amount of radioactivity in the blood at different time points, over 6 hours. While their method cannot detect free nanoparticles remaining in circulation and cells without sufficient nanoparticles to be isolated, our method does not provide details on potential interactions between nanoparticles and cells or platelets in circulation. In other words, their method can assess cellular distribution, while ours provides information on total blood clearance.

In our hands, approximately 40-50% of the injected dose of PLGA-PEG nanoparticles with short circulation times (that is, 15 PEG chains per 100 nm²) is cleared 15 minutes after injection. Provided that comparable amounts of non-crosslinked nanoworms are found in the blood after 10 minutes (as suggested in Wang et al. (2014) ACS Nano (PMID: 25419856)) and since the cell uptake method does not provide quantitative information about SPIO present in samples, it is difficult to assess what proportion of the nanomaterial is distributed within leukocytes (mostly lymphocytes) in the 10 minutes following injection. It is possible that subtle differences in cellular distribution are dependent on complement activation, but go unnoticed when measuring total radioactivity. – Since both methods offer a complementary perspective at the micro- and macroscale, respectively, it is not necessarily surprising that results differ.

2) The physicochemical properties of dextran are different from those of PEG (that is, poly(ethylene oxide)). In Inturi et al. (ACS Nano (2015), when dextran is crosslinked to obtain a poly(ethylene oxide)-containing layer, the fate of nanoworms regarding complement activation is significantly altered (complement activation in mice (not in human) is decreased, while leukocyte uptake is decreased in mice and human) – This suggests that the polymer architecture strongly influences complement activation and confirms that different mechanisms might be occurring in mice and humans.

3) In Simberg and Moghimi's work, complement-dependent uptake by murine leukocytes *in vivo* is **evidenced only with strong complement activators**. With non-crosslinked nanoworms (ACS Nano), and antibody-coated nanoworms (Front Immunol), two systems which elicit a strong complement response *in vitro* and *in vivo*, injections in WT and C3^{-/-} animals show differences in the interactions of nanoworms with blood leukocytes. These differences are not highlighted for weaker complement activators (*e.g.*, crosslinked nanoworms without Ab). In Wang *et al*, the authors acknowledge lower complement activation and decreased leukocyte uptake for nanoworms with lower surface densities of antibodies. With these systems, no data resulting from experiments in C3^{-/-} mice is presented. – Overall, it is unclear whether the fate of these mild(er) complement activators in C3^{-/-} mice was investigated in the scope of these previously published papers. Nevertheless, we believe the data presented in our paper with PLGA-PEG nanoparticles elegantly complements those obtained with nanoworms, and provides a perspective on the involvement of complement in the case of sterically-stabilized nanoparticles.

Finally, the work by Ishida and others on the “accelerated blood clearance” focuses on animals which have received sensitizing doses of nanoparticles. These animals have circulating (anti-PEG) antibodies which discernibly alter the biological response to subsequent doses of nanoparticles. As such, nanoparticles recognized by these antibodies would become strong complement activators, and the effect of the cascade on their clearance could be significantly altered.

The main text has been modified to reference to these studies.

As for the claim that the authors' results are consistent with the long circulation time of pegylated liposomes despite complement activation is not a strong argument as it can be explained by many other effects, see for example Szegeni et al., J Liposome Res 10:467, 2000.

This sentence was added to put our findings in context more than to provide an argument. To the best of our knowledge, this paradox has yet to be resolved. Various explanations have been proposed in the abovementioned publication and elsewhere: 1) iC3B cannot reach CD11b/CD18 due to PEG; 2) C3bn complexes circulate longer; 3) binding of C3bn complexes to CR1 on RBC; 4) soluble iC3b is released and blocks/saturates CD11b/CD18. The fact is that we also find PLGA-PEG nanoparticles are mild activators

of the complement cascade, while their circulation in the bloodstream seems unaffected by the complement cascade. The text has been amended to avoid confusion regarding that aspect (see above).

I would suggest that if the authors wish to maintain their conclusion about the lack of impact of C activation on their NP clearance they should provide evidence of either increased C3 binding to their NPs in wild-type vs KO mice) e.g., proteomics analysis, Suppl Fig 6, or demonstrate C activation, either by the rise of C split products in blood, or signs of C- activation-related symptoms in wild-type mice. The issue of C activation in mice is highly debated in any way, partly because of the known instability of the classical pathway in this species.

In vitro activation of the complement cascade by nanoparticles, and *in vivo* levels of C5b-9 after the injection of nanoparticles have been investigated to corroborate the activation of the complement cascade by PLGA-PEG nanoparticles. Although complement activation *in vitro* is evident, no significant rise in the production of C5b-9 complexes could be observed *in vivo*. Together, this data suggests that PEG-PLGA nanoparticles might be mild activators of the complement cascade.

Finally, although we agree that the use of mice has limitations, we still think the models have relevance to advance our understanding of interactions between nanomaterials and complete biological systems.

It is possible that the authors miss to recognize what is really interesting and important in their study, namely that PEG-PLGA NPs do not bind C3 and activate C simply because they are near neutral. This would explain the identical clearance of these NPs in mice and rats, this would be the real novelty of the paper at least regarding complement.

We thank the reviewer for this wise comment. From experiments suggested by the reviewer, we rephrased the manuscript regarding activation of the complement cascade by PEG-PLGA nanoparticles. Nevertheless, we believe one of the very interesting and novel point of our work is in showing that nanoparticles can show very different circulation times (over 7-fold changes in blood exposure), without apparent involvement of the complement cascade. This suggests that complement activation (or the lack thereof) cannot be the sole predictor of circulation times.

The text has been amended to reflect those thoughts, and highlight the limitations of studying the complement cascade in mice.

Our results suggest that complement activation **cannot be the sole** predictor of circulation times *in vivo* given that the biological fate of both short- and long-circulating nanoparticles appeared unaffected by disruption of this cascade. The biological relevance of the complex interactions between nanoparticles and this cascade of the innate immunity are still being unravelled,³⁸ **especially in mice where instability of certain activation pathways has been highlighted.**³⁵

Reviewer #3:

The authors have correctly addressed my concerns regarding the presentation and evaluation of the proteomics data. Raw data and search results have been submitted to the ProteomeXchange repository, allowing other researchers to reproduce the analyses.

The authors have correctly re-evaluated the proteomics data and the data presented are now supported by the raw data.

We thank the reviewer for his/her valuable comments and his/her time in evaluating our manuscript.

REVIEWERS' COMMENTS:

Reviewer #2 (Remarks to the Author):

The authors have thoroughly and wisely addressed my concerns, I have no more doubts about the manuscript's complement-related findings and interpretations.